# Reversible Splenial Lesion Syndrome (RESLES) after Nitrous Oxide Abuse: A Case Report

**DOI:** 10.3390/brainsci12101284

**Published:** 2022-09-23

**Authors:** Yiming Tao, Jie Han, Xiangdong Jian, Yongsheng Li

**Affiliations:** 1Department of Intensive Care Medicine, Tongji Hospital, Tongji Medical College, Huazhong University of Science and Technology, Hankou, Wuhan 430030, China; 2Department of Emergency, Tongji Hospital, Tongji Medical College, Huazhong University of Science and Technology, Hankou, Wuhan 430030, China; 3Department of Emergency, Qingdao Municipal Hospital, School of Medicine, Qingdao University, Qingdao 266071, China; 4Department of Poisoning and Occupational Diseases, Qilu Hospital of Shandong University, No. 107, Road Wenhuaxi, Jinan 250012, China

**Keywords:** reversible splenial-lesion syndrome, nitrous oxide, vitamin B12, case report

## Abstract

Background: Reversible splenial-lesion syndrome (RESLES) is a relatively rare and underrecognized clinical-imaging syndrome involving the splenium of the corpus callosum (SCC). RESLES can be caused by various etiologies. Case description: An 18-year-old man with no previous history of neurological or psychiatric disorders presented to our hospital with headache, intermittent blurred vision, and limb weakness after 150 days of recreational nitrous-oxide abuse. The patient’s serum vitamin B12 concentration was normal, and magnetic-resonance imaging (MRI) examination revealed isointensity on T1-weighted imaging (T1WI) of the corpus callosum and high signal intensity on T2WI, T2FLAIR, and diffusion-weighted MRI (DWI); thus, a diagnosis of RESLES was established. The patient received 0.5 mg of mecobalamin daily and nitrous oxide was discontinued. After 4 weeks, the patient’s symptoms disappeared and the imaging examination revealed normal findings. Conclusion: We report for the first time a case of headache, blurred vision, and hallucination caused by RESLES associated with nitrous-oxide abuse. In cases of headaches and hallucinations of unknown etiology, the possibility of RESLES caused by nitrous oxide abuse should be considered.

## 1. Introduction

Reversible splenial-lesion syndrome (RESLES) is a clinical-imaging syndrome involving the corpus callosum, which is caused by various etiologies proposed by Garcia-Monco et al. [1]. It is characterized by oval, non-enhancing lesions of the splenium of the corpus callosum (SCC) on magnetic-resonance imaging (MRI), which can completely disappear after timely and reasonable treatment. RESLES can be secondary to various diseases, including infections, epilepsy and antiepileptic drug withdrawal, autoimmune diseases, metabolic disorders, and alcohol consumption. The manifestations of RESLES are mainly related to its primary etiology, and the most common symptoms include headache, mental disturbance, altered state of consciousness, and seizures [2,3]. In addition, some patients also develop focal neurological deficits and visual symptoms [4]. Due to the rarity of the disease and lack of characteristic clinical manifestations, many clinicians do not fully recognize the disease and often misdiagnose it as a psychiatric disorder. 

## 2. Case Description

An 18-year-old man presented to our hospital with headache, intermittent blurred vision, and limb weakness after 150 days of recreational nitrous-oxide abuse (approximately 500–1000 cans of 8 g nitrous oxide per week). At the time of presentation, the patient reported limb weakness since about 90 days prior and increasing pinprick-like pain in the hands and feet. 

One week before the first medical examination, he had intermittent headaches, blurred vision, and difficulty falling asleep. Laboratory-test results showed that the serum vitamin B12 concentration was 227 pg/mL (normal range: 150–300 pg/mL), and fundoscopy was performed to exclude eye diseases. The initial local clinician diagnosed the mental symptoms caused by nitrous-oxide addiction. The patient was instructed to use olanzapine (2.5 mg/d) for treatment; however, the patient’s condition worsened after one week of treatment. On admission, the frequency of headaches and blurred vision was increased to more than 10 times a day. Headache was specifically manifested as forehead pain. The patient did not develop any nausea or vomiting, coma, or convulsions. The pain affected his sleep, but he could still fall asleep naturally. In addition to the needle-like pain at the end of the limbs, the patient was also accompanied by sensory disturbances in the lower limbs, mainly manifesting as hypoesthesia and impaired positional sense. The Mini-Mental State Examination (MMSE) score was 28, and the headache impact test-6 (HIT-6) score was 69 (Table 1). Muscle power in the proximal muscles of both upper and lower limbs was 5/5, muscle power of the ankle joint was 4/5, bipedal knuckle dorsiflexion was 3/5, plantar flexion was 3/5, and right tibialis anterior was 3/5. The physiological reflex was normal, and the pathologic reflex was not elicited. The finger-to-nose test was accurate and confirmed stability, and the Romberg test was negative. However, the patient found it difficult to walk in a straight line because of the reduced muscle strength in the right lower limb. The electroencephalogram (EEG) was normal. Electromyoneurography testing showed peripheral neuropathy in both lower limbs and spontaneous potential in the right tibialis anterior muscle; thus, the possibility of nervous-system injury was considered. MRI examination showed high signal intensity on T2-weighted imaging (T2WI) and isointensity on T1WI of the corpus callosum, and high signal intensity on T2WI, T2FLAIR, and diffusion-weighted MRI (DWI) (Figure 1); the spinal cord was normal. Laboratory tests showed the following findings: serum vitamin B12 concentration 221 pg/mL (normal range: 150–300 pg/mL), homocysteine level 28.4 μmol/L (normal range: 5–15 μmol/L), folic acid level 9.03 nmol/L (11–54 nmol/L), hemoglobin level 117 g/L (normal range: 120–160 g/L), and cerebrospinal-fluid pressure 110 mm H_2_O. Infection and autoimmunity-related examinations did not show any abnormality. The patient had started self-administration of oral vitamin B12 (500 μg/d) from the onset of limb weakness 90 days prior, which may have been responsible for the normal serum vitamin B12 concentration. 

We asked the patient to stop using nitrous oxide and olanzapine, and we started mecobalamin 0.5 mg/d (intramuscular injection) and folic acid 15 mg/d (oral administration). After 3 days of treatment, the patient reported that he had auditory hallucinations and visual hallucinations when closing his eyes. When stimulated by sound and light, the patient could switch the hallucination images involuntarily. When the patient opened his eyes, the hallucinations suddenly disappeared. After 1 week of treatment, hallucinations after eye closure disappeared, the frequency of headache and blurred-vision episodes was significantly decreased, and the self-reported difficulty falling asleep was reduced. Reperformance of the laboratory tests revealed the following findings: serum vitamin B12 concentration 216 pg/mL, homocysteine level 20.6 μmol/L, and folic acid level 16.17 nmol/L. As the patient’s condition improved, the use of mecobalamin was discontinued; it was changed to oral vitamin B12 0.1 mg daily, and the patient was asked to strictly abstain from using nitrous oxide. After 3 consecutive weeks of treatment outside the hospital, headache and blurred vision completely disappeared and needle-like pain at the tips of hands and feet was reduced as compared with that before the treatment, but complete recovery of the weakness of lower limbs was not achieved. MRI examination of the head showed disappearance of the abnormal signal in the SCC. During the 6-month follow-up period, the patient did not have any headache, intermittent blurred vision, or hallucinations, but full recovery of limb weakness was not achieved.

## 3. Discussion

Reversible splenial-lesion syndrome is a rare clinical-imaging syndrome [3], and its main causes include epilepsy, infection, vaccination, and long-term alcohol consumption. In MRI, RESLES is usually characterized by oval-shaped lesions with a clear boundary and no obvious space-occupying effect, showing equal or low signal intensity on T1WI, high signal intensity on T2WI and FLAIR, and high signal intensity on DWI [2]. To the best of our knowledge, there are no case reports of RESLES associated with nitrous-oxide abuse. Moreover, the clinical manifestations of RESLES lack specificity [3,4]; thus, RESLES is likely to be underestimated and underdiagnosed in nitrous-oxide abusers. 

It is well known that nitrous oxide can irreversibly oxidize the cobalt center of vitamin B12, causing inactivation and loss of vitamin B12, which in turn leads to impaired nucleoprotein methylation, demyelination, and axonal degeneration [5]. Based on this analysis, vitamin B12 deficiency may also cause myelination disorders and axonal degeneration in the central nervous system in the same way. In addition, vitamin B12 deficiency causes an increase in the tumor necrosis factor-alpha (TNF-α) level in the cerebrospinal fluid, which destroys the myelin and increases the blood–brain barrier permeability, further increasing myelin and interstitial edema [6]. Intramyelin or interstitial edema has been observed in animal models of subacute combined degeneration of the spinal cord [7]. Although the pathogenesis of RESLES is still not fully understood, the mainstream view is that it is more likely to be caused by edema of the myelin sheath and myelin space [1]. Therefore, based on mechanistic studies, we speculate that vitamin B12 deficiency induced by nitrous oxide may lead to RESLES by causing intramyelin and interstitial edema. In addition, although currently there is no clinical evidence for a direct causal relationship between vitamin B12 deficiency and RESLES, previous clinical studies have suggested that there may be a correlation between the two conditions [4]. From this point of view, it is necessary to further observe the changes in TNF-α, interleukin (IL)-6, IL-1, and other inflammatory indicators in vitamin B12-deficient patients, their neurological imaging changes during different time periods, and the relationship among serum vitamin B12, inflammatory indicators, and neurological imaging so as to draw more reliable conclusions. Therefore, further collection of clinical samples and investigations should be performed to provide a deeper understanding of this issue.

However, the serum vitamin B12 level in our patient was not reduced after the visit. There are numerous recent reports of nitrous-oxide abusers with normal serum vitamin B12 levels who developed symptoms associated with vitamin B12 deficiency, such as neurological symptoms, subacute combined degeneration (SCD), and megaloblastic anemia [8,9]. The main reason for normal serum vitamin B12 levels in such patients is that the current clinical-test results reflect the total serum vitamin B12 level, in which only a small proportion of biotin B12 is active and shows a dynamic change; when active vitamin B12 deficiency occurs, the total serum vitamin B12 level can still be within the normal range [10]. Capdevila et al. [11] reported that there was no linear correlation between the degree of neurological damage and serum vitamin B12 levels. In addition to the above viewpoints, it is well known that nitrous oxide can not only accelerate the excretion of vitamin B12, but also irreversibly oxidize the central cobalt part of vitamin B12 from a 1+ to a 2+ valence, thus losing its function [4]. This may be the reason why our patient continued to take vitamin B12 and the serum vitamin B12 levels remained normal, but development of functional vitamin B12 deficiency could not be avoided [12]. This phenomenon will become more common as the use of nitrous oxide becomes more prevalent and an increasing number of nitrous oxide abusers choose to self-administer vitamin B12. Therefore, the serum vitamin B12 level cannot be used as an absolute indicator to judge whether a patient has vitamin B12 deficiency. In such patients, after imaging changes in the corpus callosum are found in an MRI, the patient’s symptoms should be combined to first determine whether a diagnosis of RESLES can be established. Subsequently, laboratory tests of the patient’s cerebrospinal fluid and blood should be performed to exclude the possibility of bacterial and viral infections, metabolic disorders, and autoimmune diseases, and inquiries into the patient’s medication history, especially epilepsy drugs, should be carried out. When the diagnosis of RESLES has been established and the etiology cannot be determined, the possibility of nitrous-oxide abuse should be considered. Such patients should be asked whether they have a history of nitrous-oxide abuse and whether they have received supplementation of vitamin B12, combined with the patient’s serum vitamin B12, MMA, homocysteine levels, and clinical symptoms, to make comprehensive judgments [13]. 

Paulus et al. [12] found that in addition to anomalies of the nervous system and hematopoietic system, normal serum vitamin B12 concentrations may be found in nitrous-oxide abusers who develop isolated mental symptoms. A head MRI was not performed because the patient did not show any obvious neurological symptoms or anemia, or the patient was in the early stage of the disease and the MRI findings were not obvious. Therefore, these psychiatric symptoms may also be caused by functional vitamin B12 deficiency, and the possibility of RESLES or the development of RESLES cannot be ruled out. These patients also demonstrated a good effect after vitamin B12 supplement therapy. In addition, similar to the study by Blair C et al. [14], we also found that in patients who develop demyelinating disease after nitrous-oxide abuse, vitamin B12 supplementation to maintain normal serum levels may not provide effective prophylaxis, and the first treatment in such patients should be immediate cessation of nitrous-oxide use.

## 4. Conclusions

Here, we report for the first time a case of RESLES associated with nitrous-oxide abuse. Despite normal serum vitamin B12 concentration in our patient, based on homocysteine examination, disease history, and symptom analysis, we still believe that nitrous oxide-induced functional vitamin B12 deficiency was the real cause of RESLES, and a good therapeutic effect was achieved by discontinuing nitrous oxide and supplementing vitamin B12. We believe that a large number of these patients are missed. We believe that this case provides some insights to consider the possibility of RESLES in patients who use a large amount of nitrous oxide, and to help emergency physicians and psychiatrists manage similar symptoms in patients with nitrous-oxide abuse. 

## Figures and Tables

**Figure 1 brainsci-12-01284-f001:**
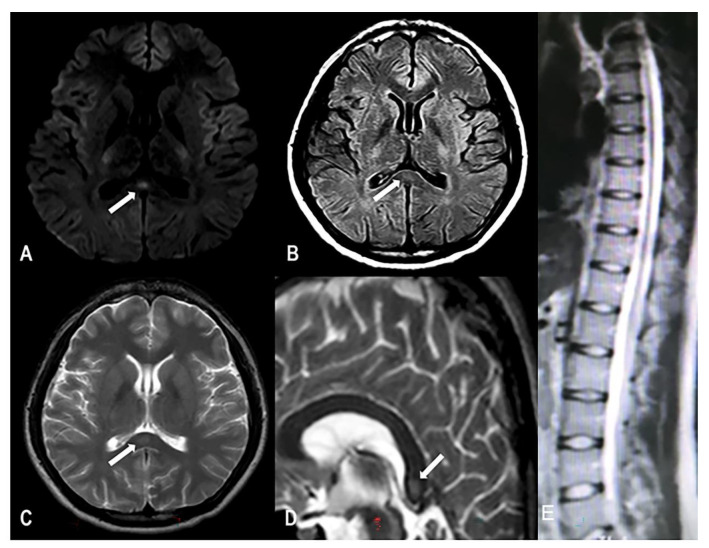
MRI demonstrating the splenial lesion. (**A**) Axial fluid-attenuated inversion recovery (FLAIR); (**B**) axial diffusion-weighted imaging (DWI); (**C**) axial T2-weighted image; (**D**) sagittal T2-weighted image; (**E**) a normal spinal cord.

**Table 1 brainsci-12-01284-t001:** Headache impact test of the patient.

Headache Impact Test				
1. When you have a headache, how often is the pain severe?				
Never	Rarely√	Sometimes	Very often	Always
2. How often do headaches limit your ability to do usual daily activities including household work, work, school, or social activities?				
Never	Rarely	Sometimes	Very often√	Always
3. When you have a headache, how often do you wish you could lie down?				
Never	Rarely	Sometimes	Very often√	Always
4. In the past 4 weeks, how often have you felt too tired to do work or daily activities because of your headaches?				
Never	Rarely	Sometimes	Very often	Always√
5. In the past 4 weeks, how often have you felt fed up or irritated because of your headaches?				
Never	Rarely	Sometimes	Very often	Always√
6. In the past 4 weeks, how often did headaches limit your ability to concentrate on work or daily activities?				
Never	Rarely	Sometimes	Very often	Always√
Never/6	Rarely/8	Sometimes/10	Very often/11	Always/13
total score				69 points

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
