# Peer review of "Reversible Splenial Lesion Syndrome (RESLES) after Nitrous Oxide Abuse: A Case Report"

_brainsci, 2022, doi:10.3390/brainsci12101284_

Round 1

Reviewer 1 Report

The authors describes a case report of reversible splenial lesion syndrome, characterized by oval, non-enhancing lesions of the splenium of the corpus callosum on MRI, after nitrous oxide abuse. Article is well introduced at the beginning and the case description seems to be detailed-described from the clinical point of view.

The article has the original topic with interesting clinical features which may be of use for the clinical practitians. At the first view, it could be of advantage to add the differential diagnostics and usefull steps for the clinitians to clearly recognise the syndrome within this perspective.    

From the format perspective, there is a need to correct almost all references in the text (extra gap after the bracket, f.e. 1st page, ref. 2-4; 4th page, ref. 6, 7, 9, 11, and 12). On page there is "Approximately" with no reason to have a capital A. Also, reference 4 does include year 2021 twice. First question in the Headache Impact Test should include "severe" option.

Author Response

Dear Reviewer 1:

Thank you very much for your comments. The article has benefited considerably from your review. We have carefully revised the manuscript according to your suggestions.  We hope that the revised parts (highlighted in red and blue) adequately address your concerns.  Thank you very much for your insightful comments, which have significantly improved our manuscript.

Point 1:

The article has the original topic with interesting clinical features which may be of use for the clinical practitians. At the first view, it could be of advantage to add the differential diagnostics and usefull steps for the clinitians to clearly recognise the syndrome within this perspective.    

Response 1:

Your comments are of great help to us. We have added the diagnostic process of reversible splenial lesion syndrome (RESLES) caused by nitrous oxide abuse, as well as the method for its differential diagnosis from other common causes (such as infection, metabolic disorders, use of anti-epileptic drugs, etc.)(Page 4,lines45-54). We believe that it is necessary to first determine whether the diagnosis of RESLES is supported by the patient's brain MRI and clinical manifestations. Diagnosis of RESLES allows other common causes to be ruled out using etiological, immunological, and metabolism-related tests. Patients were asked carefully about their history of medication (especially anti-epileptic medication) and nitrous oxide abuse. If the patient does not indicate a history of nitrous oxide abuse or the patient's serum vitamin B12 does not rule out RESLES induced by nitrous oxide abuse, the combination of the patient's serum vitamin B12, MMA, homocysteine levels, and clinical symptoms can be used to make a comprehensive judgment.

Point 2:

From the format perspective, there is a need to correct almost all references in the text (extra gap after the bracket, f.e. 1st page, ref. 2-4; 4th page, ref. 6, 7, 9, 11, and 12). On page there is "Approximately" with no reason to have a capital A. Also, reference 4 does include year 2021 twice. First question in the Headache Impact Test should include "severe" option.

Response 2:

We have corrected the references in the text, removing the extra gap after the bracket. The first letter of the word "Approximately" has been changed to lowercase(Page2 lines3). We have also revised the format of reference 4 and amended the first question in the Headache Impact Test(Page3 lines7).

We look forward to hearing from you regarding our submission. We would be glad to respond to any further questions and comments that you may have.

Reviewer 2 Report

The authors report a case of splenial lesion of the corpus callosum in a subject with nitrous oxyde chronic use.

Some changes changes could improve the reading experience.

in the title:

Rading too fast, I first thought that the authors were referring to a splenial lesion of the spleen, the abdominal organ. They shoudl add corpus callosum in the title to avoid such mistake.

I would not use the term "successful treatment" when visual hallucinations occured after the onset of methylcobalamine (suggesting a side effect) and knowing that the patient has still troubles walking "recovery of the weakness of lower limbs was not achieved".

introduction:

their is an important study of 63 patients that the authors should read and cite:

Jiang et al. Brain Behav 2021 PMID: 34758196

this could improve their initial description of the time course of the disorder and its treatment.

case description:

there is a discrepancy between the clinical description of the troubles observed in walking (normal relfexes,  the results of a peripheral or central impairment.

as no sensory deficicts are reported and the reflex are present, the authors seem to attribute the troubles in walking to the central lesion observed.

but there is no discussion with the discrepency of the EMG finding peripheral lesion.

visual hallucinations are reported only after the start of methylcobalamine. theu authors should discuss the possibility of a side effect or acknowledge that the "psychiatric symptoms" were poorly assessed at entry.

the success of the treatment is highly questionable: the evolution looks very much the same as untreated patients (see PMID: 34758196).

A more careful discussion on the effects of the treatment (positive or side effects) and the effect of time should be performed.

Author Response

Dear Reviewer 2:

Thanks very much for taking your time to review this manuscript. I really appreciate all your comments and suggestions! Please find my itemized responses in below and my revisions in the resubmitted files.

Point 1:

Rading too fast, I first thought that the authors were referring to a splenial lesion of the spleen, the abdominal organ. They shoudl add corpus callosum in the title to avoid such mistake.

 Response 1:

The question is very reasonable. The phrase splenial lesion is indeed ambiguous and easily misunderstood. However, the description “reversible splenial lesion syndrome” has been widely used since it was proposed in 2011. A search of the relevant literature showed that the terms “reversible splenial lesion syndrome” and “corpus callosum” rarely appear together in the title (we only found one article: Reversible splenial lesion syndrome: A differential diagnosis of corpus callosum lesions. PMID:32288299). I don't know if this is an effort to avoid repetition. However, your suggestion is very reasonable, and we have, therefore, decided to include the acronym for reversible splenial lesion syndrome -RESLES - in the title. RESLES is commonly used and has almost no ambiguity. I hope readers can avoid misunderstandings. We are not sure if you are satisfied with this modification? If you think this revision is inappropriate, we'd be happy to make another revision(Page 1 lines 3).

Point 2:

I would not use the term "successful treatment" when visual hallucinations occured after the onset of methylcobalamine (suggesting a side effect) and knowing that the patient has still troubles walking "recovery of the weakness of lower limbs was not achieved".

Response 2:

Thank you for the suggestion; our statement should indeed have been more rigorous. The patient's symptoms of lower extremity weakness did not fully recover, so our use of "successful treatment" was not rigorous enough, and we deleted it. However, the patient developed visual hallucinations during treatment, which we do not think is a side effect caused by methylcobalamine. On the one hand, it is possible that RESLES may cause visual hallucinations while, on the other hand,methylcobalamine has no related or similar side effects. Furthermore, with treatment, the symptoms were quickly relieved and eventually controlled. Nevertheless, as you said, "successful treatment" is really not rigorous enough, and we have revised it(Page 1 lines 2).

Point 3:

Their is an important study of 63 patients that the authors should read and cite:

Jiang et al. Brain Behav 2021 PMID: 34758196.This could improve their initial description of the time course of the disorder and its treatment.

Response 3:

Thank you so much for the tip. “Nitrous oxide-related neurological disorders: Clinical, laboratory, neuroimaging, and electrophysiological findings” is indeed an important study. We also benefited significantly from this research while reading the relevant literature and received a lot of inspiration. The study details neurological disorders associated with nitrous oxide and deserves careful reading by all researchers. However, the cases that we introduced are quite special in that they are consistent with the results of this study. For example, the spinal cord MRI did not change significantly and the limb strength decreased while at the same time, there were also some new features, such as the decrease in vitamin B12, that were not obvious, as well as discovering for the first time that RESLES can be associated with nitrous oxide. These novel features were the ones we focused on in the article and we are, therefore, concerned that citing the above study would lead to a straying from the main theme. Nevertheless, the study is indeed very important, as it forms a basis for studying these diseases and also guides our treatment work, so we would prefer to cite this publication in the Discussion section. Not sure if this modification is reasonable?(Page 4 lines 35;Page 6 lines4-5)

Point 4:

there is a discrepancy between the clinical description of the troubles observed in walking (normal relfexes,  the results of a peripheral or central impairment.

as no sensory deficicts are reported and the reflex are present, the authors seem to attribute the troubles in walking to the central lesion observed.

but there is no discussion with the discrepency of the EMG finding peripheral lesion.

Response 4:

We checked the initial medical history and confirmed that the description of the patient's muscle strength was accurate. The proximal muscle strength of the patient's lower extremities was indeed normal, but the muscle strength of the ankle joint and plantar flexors had decreased. At the same time, the muscle strength of the patient's right tibialis anterior muscle was also significantly decreased, and the muscle strength was only grade 3. While the patient had difficulty walking in a straight line, he could still qualify as "gait impairment but able to walk unsupported". We mentioned in the article that the patient experienced pin-prick pain in the extremities. At the same time, the patient's lower limbs developed sensory disturbances, mainly manifested as hypoesthesia and impaired positional sense, which we have added in the article. However, the MRI of the patient's spinal cord did not reveal abnormalities, in agreement with the results of previous studies (Nitrous oxide-related neurological disorders: Clinical, laboratory, neuroimaging, and electrophysiological findings) and suggesting the relatively poor sensitivity of MRI for the detection of N2O-induced neurological lesions. Combining these findings with the electromyography results indicates that the patient's peripheral nervous system was damaged. Therefore, we do not believe that the patient's walking difficulties originated in the central nervous system. It is well known that long-term use of nitrous oxide can cause subacute combined degeneration and peripheral neuropathy primarily involving the lower extremities. Our intention was to believe that the patient's walking difficulties and EMG changes were also caused by nitrous oxide-induced peripheral neuropathy. This does not contradict the focus on nitrous oxide-induced RESLES, and both can be present in patients. This may also explain why after treatment, the patient's central nervous system symptoms (headache, intermittent blurred vision, visual hallucinations) were rapidly relieved while the peripheral nervous system symptoms were not and did not recover easily. And the RESLES caused by nitrous oxide are what we want to talk about.However, our description of the patient's symptoms was indeed not sufficiently detailed and may cause ambiguity. We have revised it in the article, hopefully to your satisfaction(Page 2lines16-22).

Point 5:

visual hallucinations are reported only after the start of methylcobalamine. theu authors should discuss the possibility of a side effect or acknowledge that the "psychiatric symptoms" were poorly assessed at entry.

Response 5:

As we explained in response to the second question: Firstly, it cannot be ruled out that RESLES caused hallucinations; Second, the patient also stopped using nitrous oxide and olanzapine at the same time, so the possibility of withdrawal symptoms should be considered. After three days of mecobalamin, the patient's symptoms began to be relieved, and we believe that mecobalamin is less likely to cause visual hallucinations. Therefore, we evaluated this possibility using the Naranjo Algorithm Assessment:

Relative questions

score

Yes

No

Do not know

Reason

1.Are there previous conclusive reports on this reaction?

0

There have been no similar reports

2.Did the adverse event appear after the suspected drug was administered?

+2

Patient developed visual hallucinations after treatment with methylcobalamine.

3.Did the adverse reaction improve when the drug was discontinued or a specific antagonist was administered?

0

Unknown

4.Did the adverse reaction reappear when the drug was readministered?

-1

5.Are there alternative causes (other than the drug) that could on their own have caused the reaction?

-1

The possibility of visual hallucinations caused by RESLES cannot be ruled out

6.Did the reaction reappear when a placebo was given?

0

Unknown

7.Was the drug detected in the blood(or other fluids) in concentration known to be toxic?

0

Unknown

8.Was the reaction more severe when the dose was increased or less severe when the dose was decreased?

0

The patient's symptoms continued to decrease during treatment with methylcobalamine

9.Did the patient have a similar reaction to the same or similar drugs in any previous exposure?

0

no.

10.Was the adverse event confirmed by any objective evidence?

0

Total score

0

After consideration, visual hallucinations are unlikely to be caused by methylcobalamine. Of course, your suggestion is quite right. This possibility cannot be completely ruled out and we should be aware of it in future observations.

Point 6:

the success of the treatment is highly questionable: the evolution looks very much the same as untreated patients (see PMID: 34758196).

A more careful discussion on the effects of the treatment (positive or side effects) and the effect of time should be performed

Response 6:

The treatment of subacute combined degeneration of the spinal cord and peripheral nervous system injury caused by long-term heavy use of nitrous oxide is difficult, and many patients fail to fully recover motor function after long-term treatment. Complications such as incontinence, paralysis, and other serious consequences may appear. We believe that our treatment was effective for treating RESLES in this patient, as it rapidly relieved the patient's headache, blurred vision, and insomnia. However, the symptoms of lower limb weakness in the patient could have been caused by peripheral nervous system injury, and the treatment cycle for that is long. It is true that there is no effective treatment for patients who have been severely injured by prolonged and heavy use of nitrous oxide. Thus, the changes in our patients were in partial agreement with those reported in classic studies. We hope that this case provides some insights to consider the possibility of RESLES in patients who use a large amount of nitrous oxide ,and to help emergency physicians and psychiatrists manage similar symptoms in patients with nitrous oxide abuse. Of course, we will continue to monitor the patient's condition and try to collect more information.

Once again, I would like to express my sincere thanks to you. Your suggestions have been of great help and have made us sort out the structure of our article. We look forward to hearing from you regarding our submission. We would be glad to respond to any further questions and comments that you may have.

Reviewer 3 Report

we read with interest the case report by Tao et al where the authors describe a unique case of a clinical subject with  headache, blurred vision, and hallucination caused by RESLES associated with nitrous oxide abuse. 

The findings are very interesting where the authors have correlated serum vitamin B12 concentrations to  nitrous oxide abusers and its relation to develop  Reversible Splenial Lesion Syndrome (RSLS).

comments,

the work has some novelty it may show that one abused drug (nitrous oxide) may be correlated to lesions of the splenium of the corpus callosum. the work is more descriptive in nature and it would be of importance to look at some inflammatory markers tumor necrosis factor-alpha (TNF-α) level, IL-6 and IL-1 or look for markers for demyelination. 

This would fit in a section of limitation which needs to be added and would add more road map for future cases or studies that may involve more patients diagnosed with  RESLES

Author Response

Dear Reviewer 3:

Thank you very much for your comments and for pointing out the deficiencies in our article. This patient showed us that nitrous oxide abuse may lead to reversible splenial lesion syndrome, and we performed an analysis of this using published basic research and case reports. However, there has been little in-depth research on RESLES, and the information on various inflammatory indicators(TNF-α、IL-6、IL-1) and neurological imaging data can help us to understand this disease more comprehensively. Your suggestions have improved the objectivity of our paper and have pointed toward our future research directions.

Point 1:

The work is more descriptive in nature and it would be of importance to look at some inflammatory markers tumor necrosis factor-alpha (TNF-α) level, IL-6 and IL-1 or look for markers for demyelination.  This would fit in a section of limitation which needs to be added and would add more road map for future cases or studies that may involve more patients diagnosed with  RESLES.

Response 1:

Your comments are very pertinent. Regular observation of the levels of tumor necrosis factor-alpha (TNF-α), IL-6, IL-1, and other inflammatory indicators is important in these patients. In addition, at present, the diagnosis of demyelination mainly depends on imaging examination (especially MRI). Therefore, regular observation of the patient’s serum levels of vitamin B12 and inflammatory indicators, together with imaging examination results, is of great significance for research into the pathophysiology and diagnosis of such diseases. This is a limitation of our study and can only be rectified by collecting more patients and observing them over a long period of time. This is also the direction of our future work(Page 4 lines 20-26).

We look forward to hearing from you regarding our submission. We would be glad to respond to any further questions and comments that you may have.

Reviewer 4 Report

The present manuscript describes a case report of 18-year old patient with neurological disturbances after 150 days of recreational nitrous oxide abuse. It is an interesting report with a detailed description of the case and procedure as well as a properly-conducted discussion. In addition, the value of the work is increased by the table and figure with MRI demonstrating of the splenial lesion. The report was written in an understandable and correct manner in accordance with the requirements of the journal. The only thing that should be changed by the Authors is the last sentence from the introduction, it should be rather moved to conclusions and at this point add the aim of the presented manuscript.

Author Response

Dear Reviewer 4:

It is our great honor to have your affirmation. We appreciate and have incorporated your suggestions, which have certainly strengthened the manuscript.

Point 1:

The present manuscript describes a case report of 18-year old patient with neurological disturbances after 150 days of recreational nitrous oxide abuse. It is an interesting report with a detailed description of the case and procedure as well as a properly-conducted discussion. In addition, the value of the work is increased by the table and figure with MRI demonstrating of the splenial lesion. The report was written in an understandable and correct manner in accordance with the requirements of the journal. The only thing that should be changed by the Authors is the last sentence from the introduction, it should be rather moved to conclusions and at this point add the aim of the presented manuscript.

Response 1:

We have moved the last sentence of the introduction to the conclusion. This change makes the structure of our manuscript more reasonable and highlights the purpose of the paper while emphasizing its significance(Page 1 lines41-43Page 5 lines21-24).

Once again we would like to express our heartfelt thanks to you. We look forward to hearing from you regarding our submission. We would be glad to respond to any further questions and comments that you may have.

Round 2

Reviewer 2 Report

the authors adequalty adressed most of my comments and query.

minor point: they do not acknowledge the possibility that the patient improved just as the normal time-course of the deleterious effect of recovery after the lesion caused by NO, independently of the provided treatment.

in the article from Jihang et al, spontaneous improvment is also observed.

Author Response

Dear Reviewer 2:

  Thanks very much for your reply and taking your time to review this manuscript.

  It is our pleasure to be able to answer some of your questions. While answering your questions, we also reorganized the relevant knowledge and gained a lot.

Point: they do not acknowledge the possibility that the patient improved just as the normal time-course of the deleterious effect of recovery after the lesion caused by NO, independently of the provided treatment.

in the article from Jihang et al, spontaneous improvment is also observed.

 Response:  My apologies. My poor choice of words caused you to misunderstand. In fact, we do not deny that nitrous oxide-induced neurological damage will improve spontaneously over time. However, several conditions must be met, specifically, timely abstention from nitrous oxide use or that nitrous oxide was only used for a short period of time or in small amounts. If the patient has used nitrous oxide in large quantities for a long time and has suffered severe neurological damage (such as severe motor dysfunction or even paralysis and incontinence), recovery is difficult. There is also a lack of effective treatment. Currently, in the presence of serious damage, it is difficult to achieve a satisfactory cure through either self-recovery or drug treatment. Thus, numerous studies have emphasized the importance of timely withdrawal of nitrous oxide and early treatment. Our patient had abused nitrous oxide in large quantities for a long time, and the motor and sensory limb dysfunction had been present for approximately 90 days. Cases such as this, according to past experience, are difficult to treat and often have poor results. Even if the nerve damage improves spontaneously, which is a long process, the nerve function usually does not return to where it was before the use of nitrous oxide (van Amsterdam JGC, Nabben T, van den Brink W, Increasing recreational nitrous oxide use: Should we worry? A narrative review. PMID: 35678512). However, RESLES were present in our patient for only 1 week, and previous studies have also shown that with early treatment of the primary disease, recovery from RESLES is often possible. Vitamin B12 supplements have been shown to be effective. Therefore, our original intention was to remind clinicians that nitrous oxide abuse may cause RESLES, and timely treatment is important and effective. This is preferable to hoping that the nerve damage caused by nitrous oxide will improve spontaneously over time.

While we agree that nerve damage caused by nitrous oxide will recover spontaneously over time, we also wish to emphasize that for severely injured patients, this kind of recovery may take a long time, and the results may not be ideal. In addition, this is the first time that RESLES associated with nitrous oxide abuse has been observed. This disease remains largely unknown and early treatment is likely to be important for patients. Of course, this requires further investigation, which will be the focus of our further work.

Once again we would like to express our heartfelt thanks to you. We look forward to hearing from you regarding our submission. We would be glad to respond to any further questions and comments that you may have.
